# Analysis of the External and Internal Load in Professional Basketball Players

**DOI:** 10.3390/sports11100195

**Published:** 2023-10-07

**Authors:** José M. Gamonales, Víctor Hernández-Beltrán, Adrián Escudero-Tena, Sergio J. Ibáñez

**Affiliations:** 1Training Optimization and Sports Performance Research Group (GOERD), Faculty of Sport Science, University of Extremadura, 10005 Cáceres, Spain; vhernandpw@alumnos.unex.es (V.H.-B.); adescuder@alumnos.unex.es (A.E.-T.); sibanez@unex.es (S.J.I.); 2Faculty of Health Sciences, University of Francisco de Vitoria, 28223 Madrid, Spain; 3Programa de Doctorado en Educación y Tecnología, Universidad a Distancia de Madrid, 28400 Madrid, Spain

**Keywords:** training, inertial device, performance, pre-season

## Abstract

The quantification of the external load and internal load of professional players is of vital importance since it provides a great deal of information on the state of the physical condition of athletes during competition and training. The aim of the present study was to quantify the external load and internal load of the players of a first level team of the Spanish basketball league for two weeks corresponding to the pre-season 2022/2023. Seventeen load variables were analyzed and organized into kinematics external load, neuromuscular external load, and internal load. All variables were normalized to the same time unit (minute). For this purpose, all training sessions were monitored using inertial devices. The results show significant differences in the external load and internal load variables depending on the task performed and the specific position. Each type of task provokes different responses in the players, with Full Game situations producing the highest values in the kinematic external load variables (*p* < 0.05). The selection of each type of task must be adjusted to the physical and technical–tactical objective to be developed. Despite the general work carried out in the pre-season, the centers are the players who bear the greatest internal load in this period. For this reason, it is necessary to individualize the training processes from the pre-season.

## 1. Introduction

External load (EL) is understood as the work that is performed by an athlete and is quantified without considering the internal characteristics of the athlete [1]. In contrast, internal load (IL) is understood as the physiological response that is produced in the player due to sport practice [2]. Quantifying and monitoring the training loads of athletes is of vital importance to optimize the performance of athletes to the maximum. Therefore, load control allows us to identify and evaluate the performance of athletes, as well as to reduce the risk of injury or illness due to overtraining [3,4]. This load control and evaluation generates a large flow of information for the coaching staff since it will allow them to recognize the evolution of physical condition, the strategies that are being developed during training or competitions [5], and to identify the individual work ranges of the players [6] to generate a solid base of data for the correct periodization of training [6]. To reduce injuries and maladaptation to training, the coaching staff must determine and maintain an optimal connection between EL and IL values [7].

The control of training loads has been carried out in multiple invasive sports, such as football [8,9], handball [10,11], and basketball [12,13]. This allows us to determine which exercise promotes greater and better adaptations of the players to the proposed stimuli [13]. Also, since basketball is considered a cooperative–oppositional sport, high-intensity actions are interspersed with pause actions, such as rapid and short movements, accompanied by jumps [14] or impacts, with the latter being a great indicator of load since they occur in the most intense moments [15]. In the same way, the PlayerLoad (PL) is one of the most reliable variables when assessing neuromuscular IL in athletes [16], thus allowing the evolution of physical condition to be monitored and fatigue to be identified in the different tasks performed during training sessions [17].

The coach is the most responsible for the preparation and planning of the training process [18]. In addition, he/she must assume the functions of designing and carrying out the different tasks during training, since he/she is one of the most important pillars to know how training is carried out [19]. Therefore, the analysis of training tasks must be carried out based on objective, valid, and reliable data [20], with the aim of determining and evaluating the effectiveness of the sessions [21]. To this end, one of the most widely used tools is the Integral System for the Analysis of Training Tasks (SIATE), which allows the different factors that affect sports training in invasive sports to be recorded and analyzed, and which has five characteristics: universality, standardization, modularity, flexibility, and adaptability [20]. The complexity of invasive sports requires that the training tasks are designed to be as similar as possible to the real context of competition [22]. To this end, coaches must employ task constraints to promote athlete learning and increase motivation levels [23,24]. The manipulation of the organizational and formal elements of the tasks (rules, size of the field, number of players, etc.) leads to specific and challenging learning situations for the players [25].

Basketball is a sport in which collective actions predominate, both defensive and offensive. On the other hand, one of the main limitations is identifying the adaptations of each of the players before the proposed work stimuli due to the different levels of physical condition of the players, which can lead to the appearance of injuries, increased fatigue, or reduced performance [26]. Therefore, monitoring and quantifying the load will be of great importance, with the aim of knowing the players’ individual responses to the training stimuli, as well as assessing fatigue and the need for recovery to prevent injuries [27,28].

The monitoring of training tasks is very important as it allows coaches to understand the load demands that the players endure. This process is fundamental during the pre-season, as the first training sessions of the players that make up the team (new players and veteran players) are monitored after a period of inactivity. These records allow the coaches to determine the individual responses of the players and the adaptation to the new demands of the tasks. Once the players’ responses have been identified, coaches can personalize task intensities during the season. Therefore, it is necessary to know what the players’ responses are like during this first period of training.

As far as is known, there are few studies that describe the training processes of professional basketball teams. In contrast, studies have been carried out in training teams [29,30]. The study of training tasks will help in the improvement of training processes during the pre-season, since the coaching staff employ greater variability in the design of the tasks, with the aim of provoking different stimuli in their players. The study of training loads in this period has been little studied. Therefore, the general objective was to analyze the training loads of a professional basketball team during the 22/23 pre-season. Specifically, the following objectives were set: (i) to find out how the coach structures his tasks, and to identify relationships between the type of task performed and the time spent on each one; (ii) to analyze or identify the differences in the load according to the type of task given; and (iii) to analyze the differences according to the specific position.

## 2. Materials and Methods

### 2.1. Desing

This study is classified as an empirical study whose methodology is quantitative. In addition, a descriptive, associative, and differential strategy was followed [31], with the aim of quantifying the training load according to the type of task and the specific position, as well as identifying whether there are differences according to the load variables.

### 2.2. Participants

A total of 15 players (age: 25.86 ± 6.8 years, weight: 87.8 ± 7.21 kg, height: 1.97 ± 0.08 m, experience: 8.2 ± 5.11 years), belonging to a professional men’s basketball team of the highest Spanish category, participated in the study. The sample was distributed as point guard (*n* = 5), shooting guard (*n* = 2), forward (*n* = 4), power forward (*n* = 1), and center (*n* = 3). The team developed six training sessions per week with an average duration of 2 h per session. The aim of these sessions was to improve the technical and tactical skills of the players through different tasks.

In order to select the sample participants of the study, some inclusion criteria were established: (i) the player must have completed at least 90% of the training sessions developed, (ii) officially be part of the team at the beginning of the pre-season, and (iii) not to present any injuries during this period.

### 2.3. Sample

The study sample consisted of 1067 cases corresponding to each of the player’s responses in each of the tasks analyzed during two microcycles of the same pre-season. It is important to note that all training tasks, including those performed during warm-ups, were included in the descriptive analysis. In addition, to extract the maximum amount of data, the sampling frequency was set at 100 Hz, extracting the maximum amount of data in the minimum unit of time.

### 2.4. Variables

The independent variables of the present study were: (i) the type of task performed during the training sessions: Unopposed Tasks, Individual Tasks, Small-Sided Game of Numerical Equality (SSGe), Small-Sided Game of Numerical Inequality (SSGi), and Full Game [20]; and (ii) the specific position (point guard, shooting guard, forward, power forward, and center). On the other hand, for the dependent variables, different indicators of EL and IL were selected (Table 1). For subsequent analysis and comparison between groups, all variables were normalized to the same unit of time (minutes).

### 2.5. Procedure and Instruments

The data collection was carried out during the first two weeks of training corresponding to the preparation phase for the 2022/23 season. During this period, the players were monitored during the different training sessions completed in order to analyze the different tasks performed. All of the tasks designed by the coaching staff during the technical-tactical training carried out on the court were analyzed. The specific sessions of shooting training, as well as the specific sessions of strength training carried out in the gymnasium, were not analyzed. Prior to data collection, both the coaching staff and the players were informed of the possible risks and benefits of the research by means of an informed consent form. Likewise, the present study was developed under the premises established by the Declaration of Helsinki (2013). Furthermore, it was approved by the Bioethics Committee of the University of Extremadura (Registration number 233/2019).

An eight-electrode segmental monitor MC-780MA model (TANITA, Tokyo, Japan) was used to measure the weight and a rod stadiometer (SECA, Hamburg, Germany) was used to measure the height of the player in order to analyze their anthropometrical measurements.

For EL monitoring, inertial devices of the brand WIMU PRO^TM^ (RealTrack Systems, Almería, Spain) were used [32,33]. The inertial device was placed in the interscapular line using an anatomical harness (Figure 1). Likewise, for IL monitoring, the players were equipped with a Garmin^TM^ heart rate strap (Garmin Ltd., Olathe, KS, USA).

For data recording, Ultra-Wide-Band (UWB) technology was used through the location of eight antennas around the sports field (Figure 2), since it allows greater accuracy and reliability of the data collected using the Global Positioning System [34]. In addition, the ANT+ system was used to synchronize the real-time positioning of the inertial device using SVIVO^TM^ software (RealTrack Systems SL, v.2020, Almeria, Spain). Next, the SPRO^TM^ software (RealTrack Systems SL, v. 990, Almeria, Spain) was used for data extraction and processing.

### 2.6. Statistical Analysis

Criterion assumption tests were performed, showing that the data for the variables in this research did not follow a normal distribution (Kolmogorov–Smirnov) [35]. Therefore, non-parametric models were used for hypothesis testing. Subsequently, a descriptive analysis of the sample was performed to characterize the data using the mean and standard deviation. The data were also characterized in terms of percentiles to analyze the different working areas of each of the variables. Next, two-stage clustering was carried out to determine how the coach divides the tasks according to time, establishing five time ranges considering the time used for all tasks.

In the same way, an analysis was carried out to determine the relationships between the different variables through the Chi-square (X^2^) and Fisher’s Statistical Test. In addition, the strength of the associations between the variables was calculated. For this purpose, Cramer’s V coefficient (Vc) [36] was used, considering a small (<0.100), low (0.100–0.299), moderate (0.300–0.499), or high (>0.500) association [37]. Contingency tables allowed the identification of associations between variable categories through the Adjusted Standardized Residual (ASR). Residuals > 1.96 indicated significant cases [36].

Finally, the differences between the dependent variables were analyzed as a function of the type of task developed by the coach and the position of the players. In turn, through pairwise comparisons, significant differences between tasks or positions were identified. For this purpose, the nonparametric Kruskal–Wallis H test was used and the Eta square (η_p_^2^) was calculated for each analysis in order to determine the effect size, and were interpreted as η_p_^2^ < 0.01 trivial, η_p_^2^ = 0.01 to 0.06 low, η_p_^2^ = 0.06 to 0.14 moderate, and η_p_^2^ > 0.14 high [38]. Statistical analysis was performed using the Statistical Package for the Social Sciences software (version 27, 2021; IBM Corp., IBM SPSS Statistics for MAC OS, Armonk, NY, USA). Statistical significance was accepted at *p* ≤ 0.05.

## 3. Results

Table 2 shows the descriptive results of the different variables selected, as well as their counterparts weighted to the same unit of time. In addition, to represent the results obtained more accurately, percentiles are expressed.

As can be seen in Table 2, some variables present poor values for the load demands initially assumed to be made by professional basketball players. It is necessary to note that in this first descriptive analysis, all training tasks were included, including those performed during warm-up. These results show that some variables, such as accelerations/min, deceleration/min, impacts/min, or PL/min, must be taken into account in order to analyze and evaluate the player’s performance. The % Max. Heart Rate must be taken into account as this variable allows the coaching staff to identify the work threshold of each player and identify the effort developed for each player.

The cluster results showed the existence of five types of tasks regarding the duration with significant differences between them (F = 4318.44; *p* < 0.001). The groups of tasks were divided into very short duration (<6.7 min), short duration (6.8–12.25), medium duration (12.26–16.21), large duration (16.22–22.14), and very large duration (22.15–30.30).

The results obtained from the analysis of the association between task type and time show a significant value (*p* = 0.001) after the Chi-square test (X^2^ = 283.993), with a low degree of association [34].

Table 3 shows the results of the association between task types and the time cluster, which allows us to appreciate the existing relationships. In addition, unopposed tasks are performed with short (6.8–12.25 min) and medium (12.26–16.21 min) durations. Individual tasks (1x1) are performed with a very short duration (<6.7 min). Two types of duration are identified for Full Game tasks: long (16.22–22.14 min) and very long (22.15–30.30 min).

The descriptive results (mean and standard deviation), the values of the association and the level of significance (*p* < 0.05) according to the external kinematic, neuromuscular, and IL variables according to the type of task, and the differences in the load (according to the type of task) are shown below (Table 4). In the same way, the results show which variables exhibit differences depending on the variables studied. To compare the results, the variables were normalized to the minute.

Table 5 presents the results related to the analysis of the external kinematic, neuromuscular, and IL variables as a function of the specific position of the player.

## 4. Discussion

The main objective of this study was to analyze the training loads of a professional basketball team during the first two weeks of the 2022/23 pre-season. In addition, cluster analysis was carried out to identify and classify the tasks performed according to the time spent on each of them. The results show that 36% of the tasks dedicated to the Full Game have a duration greater than 16.22 min (five vs. five-game simulation conducted in half or full court, in which the coach explains and introduces the team’s attacking and defensive tactical concepts), with a higher-than-expected probability of performing these tasks with a long or very long duration. On the other hand, 91.2% of the tasks dedicated to individual work, such as technical tasks for warm-up, layup, or dribble, free throws, individual technique circuits, etc., have a short or very short duration, i.e., they do not exceed 12 min of work. The coaching staff must prepare and analyze the tasks in advance in order to quantify and establish the time and objectives of each task. During the training sessions, they must determine whether they meet the objectives or if they must change the dynamics of the task. In the same way, they must not exceed the time for each task and should develop different tactical and technical skills.

Depending on the type of task, significant differences can be observed in all of the load variables analyzed. In the same way, when analyzing the differences between pairs: Without opposition—Full Game; Without opposition—Equality SSG; Without opposition—Individuals, those that present differences in all of the EL and IL variables were analyzed. The Full Game tasks are the ones in which the highest values have been identified in the IL variables, as well as in the absolute variables related to kinematic EL. This is because in a match simulation where five vs. five is played, higher values are obtained than in the rest of the situations since it is a simulation of a maximum-intensity scenario [22]. On the other hand, the individual game situations present higher values in the neuromuscular variables, such as PL, impacts, or the number of jumps, due to being tasks that are carried out in high-intensity individual offensive game situations (counterattacks, fastball starts), and circuits to work on individual strength or technical skill. Therefore, the different game situations should be considered when planning the tasks, as they will allow for better organization and optimization of the sessions [39] through the manipulation of the organizational and formal elements of the tasks (rules, size of the field, number of players, etc.) [23,24,25]. The coaching staff are the most responsible for the preparation and planning of the training process [18]. They must carry out a process of preparation and design of the training tasks according to the needs and objectives proposed, since a variation in the playing space or the time spent will have a direct impact on the load perceived by the player, with full game tasks being those that produce the greatest fatigue in athletes [39]. Also, they must evaluate the training sessions and determine whether the objectives were developed in the best way and assess the effectiveness of the sessions [21]. Equality SSG situations are the most physically demanding for players [40,41]. In the same way, exercises that replicate the intermittency of effort during competitions should be carried out, with the aim of performing tasks in training with the same pattern of effort as in competition. In the pre-season, the design of the tasks and the order during the session are fundamental to achieve the physical and technical–tactical objectives. This period of training must be used to increase the training load and adjust the periodization [42]. The coaching staff must evaluate the differences between the players and personalize the training sessions regarding the style of play and player personnel [43]. This period will allow the players to acquire the appropriate fitness condition for the in-season.

Finally, depending on the specific position, significant differences are observed in most of the EL and IL variables, except for distance/min, HSR/min, and average speed. In addition, it is observed that it is the Centers who have the highest values in the IL variables, since being the inside players, they are the ones who receive the highest number of jumps and impacts, and, therefore, the highest PL values. These variables are the most representative for determining the load submitted by the athletes [15]. These results are in line with those obtained by Ibáñez, Piñar, García, and Mancha-Triguero [40], since each specific position presents some characteristic physical demands, and, therefore, there are differences in the different EL and IL variables analyzed. Therefore, due to the disparity in the results obtained depending on the position, the principle of individualization of training must be considered, with the aim of personalizing training sessions considering the work thresholds of each player [44]. This personalized monitoring and evaluation of the training load will reduce the risk of injury, as well as allowing an exhaustive and systematic control of the evolution of the players’ physical condition [45]. Therefore, monitoring and quantifying the load will be of great importance in understanding and analyzing the individual responses of each training session, as well as in assessing fatigue and recovery time to prevent injuries [27,28]. Due to the importance of understanding the physical demands of the players, the coaching staff must determine the limits of each of the athletes in order to develop personalized training sessions adapted to the needs of each of them. Also, due to the control of the training loads, it is necessary to know how the stimuli produced in the training sessions influence the players and determine whether the players adapt correctly to them.

The coaching staff must be aware of the physical demands to which their players are subjected in order to analyze and evaluate the progression in physical condition, and also to avoid the generation of overload injuries. One of the limitations of the study is that the sample corresponds to a single top-level professional team [40], so the results cannot be extrapolated to the entire population of basketball players. Also, another limitation is that the sample was small, and it would be useful in the future to attempt to increase the number of players and teams assessed as well as extending the duration of the analyses. In this way, the results can be extrapolated to other teams. The present study is one of the first to provide information and analyze the load borne by professional basketball players, making it possible to provide reference values for other groups.

## 5. Conclusions

Quantifying training loads will allow coaching staff to objectively assess the performance of players during training sessions or in competitive periods. This will allow them to adapt the stimuli of the training sessions to the demands produced during competitions so that the loads reflect those of a real game situation. Moreover, systematic measurement will make it possible to obtain relevant data on the strategies developed in the training sessions and to control the evolution of physical performance, as well as the appearance of injuries or fatigue.

The monitoring of players during the pre-season makes it possible to follow the first training sessions of the players who make up the team after a period of inactivity. These records allow the coaches to monitor the individual adaptations of the players to the demands of the tasks and to personalize the intensities of the tasks during the season.

Depending on the objectives proposed by the coaching staff, one type of task or another must be developed, as each of them will produce different physical demands on the players. Therefore, Equality SSG situations should be worked on for the development of collective technical–tactical skills. On the other hand, Full Game situations produce the highest values in the kinematic EL variables, as they are situations that are like the physical demands of competition. At the same time, personalized training should be carried out according to the characteristics of the players and the specific position, since each position has its own game characteristics.

This information is of great interest to the scientific field, and especially for basketball, as it will increase the knowledge regarding the technical–tactical behavior of the players depending on the type of task and in real-game simulation situations, as well as in determining the influence of specific positions on the EL and IL values.

## Figures and Tables

**Figure 1 sports-11-00195-f001:**
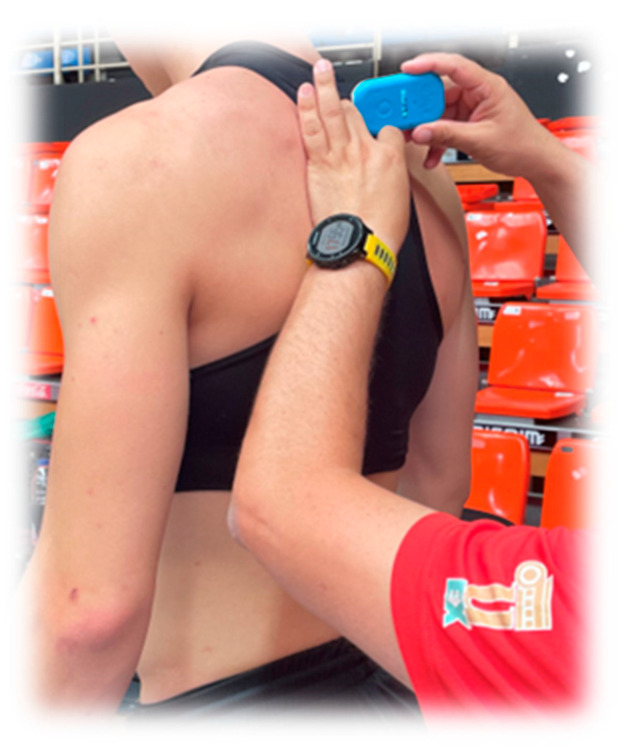
Positioning of the ID in the interscapular area by means of an anatomical harness.

**Figure 2 sports-11-00195-f002:**
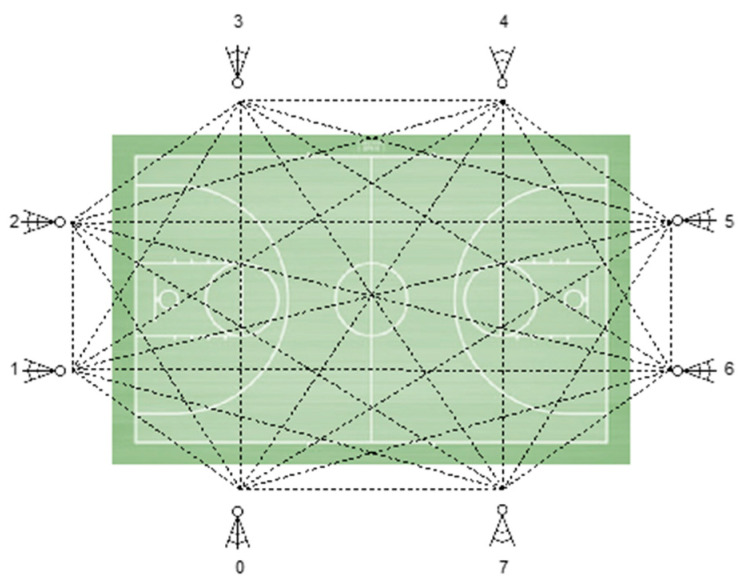
Distribution of antennas on the court.

**Table 1 sports-11-00195-t001:** Description of selected dependent variables.

Variable		Unit	Description
Kinematics EL	Distance	Meters (m)	Space covered
	Explosive Distance	Meters (m)	Space covered with higher acceleration of 1.12 m/s^2^
	High Speed Running	Meters (m)	Distance covered over 21 km/h
	Acceleration	Number (n)	Total positive speed change
	Deceleration	Number (n)	Total negative speed change
	Max. Acceleration	m/s^2^	Maximum capacity of increased speed
	Max. Deceleration	m/s^2^	Maximum capacity of slowdown
	Average Speed	km/h	Average speed
	Max. Speed	km/h	Maximum speed
Neuromuscular EL	PL	Arbitrary unit (a.u.)	Cumulative load in relation to accelerations on the 3 axes
	Impact	Number (n)	Number of impacts during the session
	Average Takeoff (G)	G force (G)	G force attained during the propulsive phase of a jump
	Average Landing (G)	G force (G)	G force attained during the land phase of a jump
	Jumps	Number (n)	Counter of the number of jumps
Objective IL	Average Heart Rate	Beats per minute (bpm)	Arithmetic mean of the number of beats in a time interval
	Max. Heart Rate	Beats per minute (bpm)	Maximum heart rate achieved
	% Max. Heart Rate	Beats per minute (bpm)	Work zones according to the percentage of the player’s maximum heart rate, Z1 (50–60%), Z2 (60–70%), Z3 (70–80%), Z4 (80–90%), Z5 (90–95%) and Z6 (>95%).

EL: external load; IL: internal load; PL: PlayerLoad.

**Table 2 sports-11-00195-t002:** Descriptive analysis and percentiles of the variables.

Variables		$\bar{X}$	SD	P.05	P.25	P.50	P.75	P.95
KEL	Distance (m)	579.99	375.50	44.48	268.45	507.01	891.32	1212.31
	Dist./min (m/min)	43.26	24.76	3.60	26.39	41.02	58.37	87.02
	Explosive Dist. (m)	54.97	46.24	0.03	16.48	45.53	86.29	144.31
	Explosive Dist./min (m/min)	3.79	2.59	0.00	1.71	3.81	5.48	8.08
	Accelerations (count)	282.09	158.85	77.40	157.00	267.00	369.00	573.00
	Accelerations/min (count)	212.70	146.00	23.80	109.00	184.00	279.00	495.60
	Decelerations (count)	20.31	6.24	6.45	17.73	21.67	24.58	28.08
	Decelerations/min (count)	15.29	6.83	2.39	10.81	15.46	20.48	25.77
	HSR (m)	14.22	30.46	0.00	0.00	0.00	13.40	80.89
	HSR/min (m)	0.94	1.98	0.00	0.00	0.00	1.00	5.04
	Max. Acceleration (m/s^2^)	3.73	1.28	1.14	3.09	3.93	4.56	5.42
	Max. Deceleration (m/s^2^)	−3.39	1.33	−5.26	−4.25	−3.53	−2.64	−0.88
	Avg. Speed (km/h)	3.86	1.17	1.86	3.07	3.85	4.69	5.73
	Max. Speed (km/h)	15.47	5.26	4.77	12.47	16.24	19.35	22.66
NEL	PL (a.u.)	8.25	6.58	0.49	2.55	7.10	12.49	20.36
	PL/min (a.u.)	1.40	2.72	0.09	0.46	0.69	1.09	8.21
	Impacts (count)	1117.39	807.51	26.40	497.00	986.00	1702.00	2538.40
	Impacts/min (count)	84.67	59.47	2.27	48.02	73.88	108.28	207.14
	Avg. Takeoff (G) (count)	2.26	1.65	0.00	1.15	2.35	3.27	4.83
	Avg. Landing (G) (count)	4.07	2.53	0.00	3.29	4.57	5.60	7.68
	Jumps (count)	4.31	4.78	0.00	1.00	3.00	6.00	14.00
	Jumps/min (count)	0.33	0.40	0.00	0.07	0.22	0.44	1.02
OIL	Avg. Heart Rate (bpm)	123.09	25.31	79.00	111.00	127.00	140.00	157.00
	Max. Heart Rate (bpm)	148.78	30.39	92.40	137.00	156.00	170.00	183.60
	% Max. Heart Rate (%)	66.63	13.61	42.88	60.20	69.10	75.30	85.40

KEL: kinematics external load variables; NEL: neuromuscular external load variables; OIL: objective internal load variable; Dist.: distance; HSR: high-speed running; PL: PlayerLoad.

**Table 3 sports-11-00195-t003:** Descriptive analysis regarding the type of task.

			Time Duration of The Task					Total
			Very Short < 6.7 Min	Short 6.8–12.25 Min	Medium 12.26–16.21 Min	Long 16.22–22.14 Min	Very Long 22.15–30.30 Min	
Type of task	Without opposition	n	28	71	43	0	0	142
		%	19.7%	50.0%	30.3%	0.0%	0.0%	100.0%
		ASR	−1.4	3.4	2.3	−4.0	−3.4	
	Individuals	n	84	71	15	0	0	170
		%	49.4%	41.8%	8.8%	0.0%	0.0%	100.0%
		ASR	8.3	1.4	−4.7	−4.5	−3.8	
	Equality SSG	n	104	154	86	14	14	372
		%	28.0%	41.4%	23.1%	3.8%	3.8%	100.0%
		ASR	1.9	2.1	0.2	−4.4	−2.8	
	Full Game	n	45	100	98	83	57	383
		%	11.7%	26.1%	25.6%	21.7%	14.9%	100.0%
		ASR	−7.2	−5.6	1.7	10.7	8.1	

n: sample; %: percentage of cases; SSG: Small-Sided Game; ASR: adjusted standardized residual.

**Table 4 sports-11-00195-t004:** Descriptive results and differences regarding the type of task.

Variables		Without Opposition		Individuals		Equality SSG		Full Game		H	*p*	Post Hoc	Eta Square
		$\bar{X}$	SD	$\bar{X}$	SD	$\bar{X}$	SD	$\bar{X}$	SD				
KEL	Dist./min	7.56	5.49	82.70	11.19	42.97	15.47	39.28	15.18	648.213	0.000	a, b, c, e, f	0.607
	Explosive dist./min	0.13	0.23	5.03	1.96	4.31	2.48	4.09	2.24	376.923	0.000	a, b, c, e, f	0.352
	Acceleration/min	73.75	72.84	173.11	69.09	220.12	122.04	274.57	170.83	264.149	0.000	a, b, c, d, e, f	0.246
	Deceleration/min	5.82	4.95	18.92	4.18	17.01	5.84	15.53	6.18	294.819	0.000	a, b, c, d, e, f	0.275
	HSR/min	0.00	0.00	0.52	1.08	1.29	2.59	1.13	1.85	97.876	0.000	a, b, c, d, e	0.089
	Max. Acceleration	1.54	0.77	3.99	1.08	3.99	0.98	4.18	0.93	334.515	0.000	a, b, c, d	0.312
	Max. Deceleration	−1.26	0.75	−3.52	1.21	−3.65	1.03	−3.88	1.03	333.022	0.000	a, b, c, d, e	0.31
	Avg. Speed	2.13	0.49	5.50	0.61	3.90	0.80	3.75	0.79	596.094	0.000	a, b, c, e, f	0.558
	Max. Speed	6.53	3.14	16.61	2.48	16.37	4.11	17.40	4.42	338.406	0.000	a, b, c, d	0.316
NEL	PL/min	0.24	0.41	2.15	2.64	1.22	1.99	1.69	3.58	437.723	0.000	a, b, c, e, f	0.409
	Impacts/min	8.63	8.42	195.15	30.30	74.69	28.55	73.52	29.03	661.139	0.000	a, b, c, e, f	0.619
	Avg. Takeoff (G)	0.15	0.93	2.06	0.86	2.62	1.54	2.78	1.62	307.028	0.000	a, b, c, d, f	0.286
	Avg. Landing (G)	0.25	0.96	3.98	1.40	4.86	2.40	4.77	2.12	319.588	0.000	a, b, c, e, f	0.298
	Jumps/min	0.01	0.03	0.70	0.67	0.33	0.30	0.27	0.23	339.774	0.000	a, b, c, e, f	0.317
OIL	Avg. Heart Rate	86.01	14.54	125.09	16.47	129.62	19.88	129.61	24.67	329.373	0.000	a, b, c, e, f	0.307
	Max. Heart Rate	101.08	16.92	147.82	18.90	156.31	22.00	159.58	28.73	380.349	0.000	a, b, c, d, e, f	0.355
	% Max. Heart Rate	46.58	7.41	67.92	8.80	69.88	10.55	70.33	13.42	326.811	0.000	a, b, c, e	0.305

KEL: kinematics external load variables; NEL: neuromuscular external load variables; OIL: objective internal load variable; Dist.: distance; Avg.: average; HSR: high-speed running; PL: PlayerLoad; SSG: Small-Sided Game; H: Kruskal–Wallis H test; *p* < 0.05; a: Without opposition—Full Game; b: Without opposition—Equality SSG; c: Without opposition—Individuals; d: Full Game—Equality SSG; e: Full Game—Individuals; f: Equality SSG—Individuals.

**Table 5 sports-11-00195-t005:** Descriptive results and differences regarding the game position.

Variables		Point Guard		Shooting Guard		Forward		Power Forward		Center		H	*p*	Post Hoc	Eta Square
		$\bar{X}$	SD	$\bar{X}$	SD	$\bar{X}$	SD	$\bar{X}$	SD	$\bar{X}$	SD				
KEL	Dist./min	43.59	24.79	42.39	24.57	42.26	24.54	44.60	24.57	44.41	25.58	1.572	0.814		0.002
	Explosive dist./min	3.94	2.55	4.06	2.61	3.59	2.52	2.60	1.85	4.21	2.94	26.521	0.000	a, b, c, d	0.021
	Acceleration/min	193.92	139.83	226.46	150.95	217.64	139.75	231.67	152.11	222.91	159.43	14.852	0.005	h	0.01
	Deceleration/min	13.91	6.18	16.34	7.11	15.85	6.93	16.42	6.97	15.80	7.26	27.458	0.000	b, e, h, i	0.022
	HSR/min	0.77	1.67	0.91	1.86	1.03	1.97	0.89	2.14	1.18	2.59	4.05	0.399		0
	Max. Acceleration	3.79	1.28	3.85	1.28	3.70	1.29	3.38	1.37	3.75	1.23	21.679	0.000	a, b, c, d	0.017
	Max. Deceleration	−3.53	1.38	−3.57	1.36	−3.35	1.29	−2.83	1.25	−3.29	1.22	36.623	0.000	a, b, c, d	0.031
	Avg. Speed	3.98	1.18	3.88	1.17	3.79	1.15	3.68	1.08	3.83	1.21	8.606	0.072		0.004
	Max. Speed	15.96	5.19	16.10	5.14	15.60	5.35	13.53	4.90	14.62	5.25	25.873	0.000	a, b, c, i, j	0.021
NEL	PL/min	1.38	2.53	1.48	2.68	1.27	2.41	1.12	2.42	1.81	3.68	23.627	0.000		0.018
	Impacts/min	84.56	60.32	81.78	59.81	81.02	57.40	77.67	59.75	97.98	59.55	17.689	0.001	d, g, i, j	0.013
	Avg. Takeoff (G)	2.27	1.70	2.45	1.71	2.12	1.60	1.78	1.47	2.60	1.61	24.525	0.000	c, d, g, i	0.019
	Avg. Landing (G)	3.82	2.26	4.78	2.97	3.79	2.38	3.42	2.39	4.86	2.70	50.333	0.000	c, d, f, g, h, i,	0.044
	Jumps/min	0.33	0.45	0.30	0.27	0.26	0.27	0.20	0.23	0.52	0.53	45.395	0.000	c, d, g, i, j	0.039
OIL	Avg. Heart Rate	123.69	27.60	122.72	25.43	122.85	23.85	112.12	19.80	128.48	23.38	38.732	0.000	a, b, c, d	0.033
	Max. Heart Rate	149.13	33.02	150.28	31.86	150.47	29.55	138.22	24.91	149.30	26.00	25.679	0.000	a, b, c, d	0.02
	% Max. Heart Rate	65.55	14.26	66.79	14.16	66.11	12.70	64.08	12.91	71.22	12.67	33.908	0.000	d, g, i, j	0.028

KEL: kinematics external load variables; NEL: neuromuscular external load variables; OIL: objective internal load variable; Dist.: distance; Avg.: average; HSR: high-speed running; PL: PlayerLoad H: Kruskal–Wallis H test; df: differences; *p* < 0.05; a: Power Forward–Forward; b: Power Forward–Point Guard; c: Power Forward–Shooting Guard; d: Power Forward–Center; e: Forward–Point Guard; f: Forward–Shooting Guard; g: Forward–Center; h: Point Guard–Shooting Guard; i: Point Guard–Center; j: Shooting guard–Center.

## Data Availability

The data that support the findings of this study are available from the corresponding author upon reasonable request (martingamonales@unex.es).

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
