# Peer review of "Analysis of the External and Internal Load in Professional Basketball Players"

_sports, 2023, doi:10.3390/sports11100195_

Round 1
Reviewer 1 Report
First, I would like to recognize the authors for the preparation of this article.
The title is precise and accurate.
Abstract
The abstract presents the rationale of the study. In addition, it provides an objective representation of the article and presents the study’s results and significance without exaggerating. Line 17: no need to capitalize full-game situations. Line 20: No need to capitalize pivots or internal load. Also, it would have been nice to present which variables were tested but based on the fact that you have a word limit it is ok not to include those in the abstract.
Introduction
The introduction is clear and follows a logical sequence while all the relevant scientific support is provided. The need of monitoring and quantifying the training load is well justified.
The purpose of the study is clearly stated.
Line 44: player load and not PlayerLoad. Be consistent in the spacing as you presented the training load with a space before.
Line 71: I would only say studies and not include references (as essentially you are referring to the same thing).
The materials and methods section is presented with sufficient detail so that someone can replicate and build on the published results. In addition, the methods were appropriately presented and cited. The results including the tables are straightforward and nicely presented.
The discussion is excellent with the limitation being presented. Concussions are clear and the overall importance of the study is well justified.
Author Response
Reviewer 1
Open Review
( ) I would not like to sign my review report
(x) I would like to sign my review report
Quality of English Language
( ) I am not qualified to assess the quality of English in this paper
( ) English very difficult to understand/incomprehensible
( ) Extensive editing of English language required
( ) Moderate editing of English language required
( ) Minor editing of English language required
(x) English language fine. No issues detected
|
Yes |
Can be improved |
Must be improved |
Not applicable |
|
|
Does the introduction provide sufficient background and include all relevant references? |
(x) |
( ) |
( ) |
( ) |
|
Are all the cited references relevant to the research? |
(x) |
( ) |
( ) |
( ) |
|
Is the research design appropriate? |
(x) |
( ) |
( ) |
( ) |
|
Are the methods adequately described? |
( ) |
( ) |
( ) |
( ) |
|
Are the results clearly presented? |
(x) |
( ) |
( ) |
( ) |
|
Are the conclusions supported by the results? |
(x) |
( ) |
( ) |
( ) |
Comments and Suggestions for Authors
First, I would like to recognize the authors for the preparation of this article.
The title is precise and accurate.
Authors: Thank you for your perspective. However, the title has been changed due to the petition of another reviewer.
Abstract
The abstract presents the rationale of the study. In addition, it provides an objective representation of the article and presents the study’s results and significance without exaggerating. Line 17: no need to capitalize full-game situations. Line 20: No need to capitalize pivots or internal load. Also, it would have been nice to present which variables were tested but based on the fact that you have a word limit it is ok not to include those in the abstract.
Authors: Thank you for your perspective. Those mistakes have been corrected.
Introduction
The introduction is clear and follows a logical sequence while all the relevant scientific support is provided. The need of monitoring and quantifying the training load is well justified.
Authors: Thank you for your perspective.
The purpose of the study is clearly stated.
Authors: Thank you for your perspective.
Line 44: player load and not PlayerLoad. Be consistent in the spacing as you presented the training load with a space before.
Authors: Thank you for your comments. This mistake has been corrected in the whole document. Also, the authors have modified the way they present the term, without space, and using the abbreviation
Line 71: I would only say studies and not include references (as essentially you are referring to the same thing).
Authors: Thank you for your note.
The materials and methods section is presented with sufficient detail so that someone can replicate and build on the published results. In addition, the methods were appropriately presented and cited. The results including the tables are straightforward and nicely presented.
Authors: Thank you for your comments. We really appreciate it.
The discussion is excellent with the limitation being presented. Concussions are clear and the overall importance of the study is well justified.
Authors: Thank you for your perspective.
Submission Date
12 July 2023
Date of this review
26 Jul 2023 13:05:31

Reviewer 2 Report
Dear Authors
You have written an interesting paper focusing on quantifying the external
and internal load of the players of a first-level team in the Spanish basketball league.
However, Several parts need to be addressed for greater clarity and reproducibility of your research.
The title should be rewritten as it si to long and hard to read.
Abstract: report significant results with p values.
Introduction:
The introduction is solid and presents the main rationale well
Methods
Participants - what was their training experience, and how many per position? report
How did you determine the sample size (G*Power software or any other method)? report / What is the power of your study? report the post-hoc results of G*Power
How did you measure body weight and height? report
Two-preseason microcycles - and..?? How many training sessions were recorded, and how long were they? What was the main aim of these sessions? Are results from all sessions - summarised or only from one session? report as it is not clear
Was also the warm-up and cool-down a part of the measurements?
Specific position - so where is the center position? elaborate / small forward stated twice - delete
small-sided game repeats twice - delete /
types of tasks - how were they defined - according to which papers?
Statistical analysis - what was your significance value? report it
Results - From Table 2 there are some unusual data:
-distance / very large SD - Why?
- extremely low number of jumps with a SD greater than the mean? Why - what kind of a training was that?
-very low heart rate
Discussion
The discussion is poorly presented and does not connect well to the used references in the introduction.
You are talking about objectives but you didn't present the objectives of this trainings and the session results don't present any added value. To state that ''The coaching staff must know how to carry out the training process'' is like stating the obvious as the team was a professional team.
The limitations section is very short and should be extended.
Overall, in this condition the methods are poorly described and some results does not make sense. Therefore, I am rejecting this study.
Extensive editing of the English language required
Author Response
Reviewer 2
Open Review
(x) I would not like to sign my review report
( ) I would like to sign my review report
Quality of English Language
( ) I am not qualified to assess the quality of English in this paper
( ) English very difficult to understand/incomprehensible
(x) Extensive editing of English language required
( ) Moderate editing of English language required
( ) Minor editing of English language required
( ) English language fine. No issues detected
|
Yes |
Can be improved |
Must be improved |
Not applicable |
|
|
Does the introduction provide sufficient background and include all relevant references? |
( ) |
(x) |
( ) |
( ) |
|
Are all the cited references relevant to the research? |
( ) |
(x) |
( ) |
( ) |
|
Is the research design appropriate? |
( ) |
( ) |
(x) |
( ) |
|
Are the methods adequately described? |
( ) |
( ) |
(x) |
( ) |
|
Are the results clearly presented? |
( ) |
( ) |
(x) |
( ) |
|
Are the conclusions supported by the results? |
( ) |
( ) |
(x) |
( ) |
Comments and Suggestions for Authors
Authors: I and my fellow authors would like to thank you for revising this manuscript, as well as for the comments. We believe we have adequately addressed the suggestions, however, if you deem that more changes are necessary, we look forward to addressing any other concerns.
Dear Authors
You have written an interesting paper focusing on quantifying the external and internal load of the players of a first-level team in the Spanish basketball league.
However, Several parts need to be addressed for greater clarity and reproducibility of your research.
The title should be rewritten as it si to long and hard to read.
Authors: Thank you for your perspective. The title has been changed and reduce the number of words in order to improve its reading.
“Analysis of the external and internal load in professional basketball players”
Abstract: report significant results with p values.
Authors: Thank you for your perspective. The p-value of the results was added.
Introduction:
The introduction is solid and presents the main rationale well
Authors: Thank you for your perspective.
Methods
Participants - what was their training experience, and how many per position? Report
Authors: Thank you for your comments.
This information has been added in participants section.
“The sample was distributed as point guard (n=5), shooting guard (n=2), forward (n=4), power forward (n=1), and center (n=3). The team developed 6 training sessions per week with an average duration of 2h per session. The aim of these sessions was to im-prove technical and tactical skills of the players through different tasks.
In order to select the sample participants of the study, some inclusion criteria were established: i) the player must have completed at least 90% of the training sessions developed, ii) officially be part of the team at the beginning of the pre-season, and iii) not to present any injuries during this period.”
How did you determine the sample size (G*Power software or any other method)? report / What is the power of your study? report the post-hoc results of G*Power
Authors: The authors thank you for your comments.
The sample size was not calculated, since this was an ecological study, all the training sessions carried out during the period of time authorized by the team were collected. No tasks were excluded from monitored sessions.
The mathematical model used to calculate the relationship between the variables was nonparametric, using the Chi-square test (X2) and Fisher's statistical test. The Cramer's V coefficient (Vc) was used to calculate the strength of association between the variables.
The degree of association, or power, between the variables was determined from the proposal made by Crewson (2006) based on the values of Cramer's V coefficient (Vc). Therefore, it was not calculated with any statistical program.
The calculation of the post-hoc tests of the nonparametric Kruskal-Wallis H test is performed automatically with the statistical program for data analysis.
The SPSS program was used for all calculations.
How did you measure body weight and height? Report
Authors: Thank you for your note.
Information was added in the Procedure and instruments section.
“An 8-electrode segmental monitor MC-780MA model (TANITA, Tokyo, Japan) was used to measure the weight and a rod stadiometer (SECA, Hamburg, Germany) was used to measure the height of the player, in order to analyse their anthropomet-rical measurements.”
Two-preseason microcycles - and..?? How many training sessions were recorded, and how long were they? What was the main aim of these sessions? Are results from all sessions - summarised or only from one session? report as it is not clear
Authors: Thank you for your comments. We apologize that our judgment has caused confusion to the reviewer. We have clarified the expression in the text by improving the sentence:
“The study sample consisted of 1067 cases corresponding to each of the player's responses in each of the tasks analyzed during two microcycles of the same preseason.”
As it is explained in the method section, a total of 12 sessions were recorded during the microcycle. The purpose of the sessions was to improve the technical and tactical skills of the players through individual tasks, or full-game tasks. The data collected and analyzed were results of all sessions.
Was also the warm-up and cool-down a part of the measurements?
Authors: Thank you for your note. In this case, warm-up and cool-down parts were not deleted, in order to analysed the load in the whole tasks of the training sessions.
A sentence has been included to clarify your suggestion.
“All the tasks performed during the training sessions were included, also warm-up and cool down, to subsequently classify them according to the type of task.”
Specific position - so where is the center position? elaborate / small forward stated twice – delete
Authors: Thank you for your perspective. Specific positions have been updated and corrected the mistakes.
Small-sided game repeats twice - delete /
Authors: Thank you for your comments. This term is not repeated, two different situations are shown, Unequal and Equal Small Sided Gamed.
The independent variables of the present study were: i) the type of task performed during the training sessions: Unopposed Tasks, Individual Tasks, Small Sided Game in Unequal, Small Sided Game in Equal and Full Game [20];
types of tasks - how were they defined - according to which papers?
Authors: Thank you for your note.
In the third paragraph of the introduction, reference is made to the SIATE tool, which is the one used to define the training tasks.
A quote has been added. It is the first study in which Those situations were described and identified. Also, some references which carried out similar analyzed were added.
- Ibáñez, S.J.; Feu, S.; Cañadas, M. Integral Analysis System of Training Tasks, SIATE, in Invasion Games. e-balonmano com 2016, 12, 3–30.
- Cañadas, M.; Parejo, I.; Ibáñez, S.J.; García-Rubio, J.; Feu, S. Relationship between the Pedagogical Variables of Coaching a Mini-Basketball Team. Rev. Psicol. del Deport. 2009, 18, 319–323.
- Ibáñez, S.J.; Pérez-Goye, E.; García-Rubio, J.; Courel-Ibáñez, J. Effects of Task Constraints on Training Workload in Elite Women’s Soccer. Int. J. Sports Sci. Coach. 2020, 15, 99–107, doi:10.1177/1747954119891158.
- Cañadas, M.; Ibáñez, S.J.; García-Rubio, J.; Parejo, I.; Feu, S. Game Situations in Youth Basketball Practices. Rev. Int. Med. y Ciencias la Act. Física y el Deport. 2013, 13, 41–54.
- Cañadas, M.; Ibáñez, S.J. Planning the Contents of Training in Early Age Basketball Teams. E-balonmano com 2010, 6, 49–65.
- Cañadas, M.; Ibáñez, S.J.; García-Rubio, J.; Parejo, I.; Feu, S. Study of the Phases of Game through the Analysis of Sport Training in U’12 Categories. Cuad. Piscología del Deport. 2013, 12, 73–82.
- Caceres-Sanchez, L.; Escudero-Tena, A.; Fernandez-Cortes, J.; Ibañez, S.J. Analysis of Training Variables of Basketball in a Formative Stage. A Case Study. e-balonmano com 2021, 17, 135–144.
Statistical analysis - what was your significance value? report it
Authors: Thank you for your perspective.
The following sentence has been included in the text:
“The significance value was identified and established in p<0.05.”
Results - From Table 2 there are some unusual data:
-distance / very large SD - Why?
Authors: Thank you for your note. The reason is that some players made less distance than the rest due to their position or because of some constraints in the task.
Moreover, since all the tasks performed, including warm-up, are analyzed in this type of task, the distance covered is significantly lower than in other tasks. On occasion, the coach determined to have a player participate less in an assignment.
- extremely low number of jumps with a SD greater than the mean? Why - what kind of a training was that?
Authors: Thank you for your comments.
The reason is that some players made less jumps than the rest due to their position or because of some constraints in the task. The coaching staff developed tasks to improve technical and tactical skills, so a great number of tasks were individual or without opposition.
When all tasks are presented descriptively, they include tasks in which the players did not perform any jumps, with others in which they performed more jumps. The scientific focused on analyzing the data extracted from the training developed by the coaches. They don’t manipulate any variable. So, the values are related to the task they made. This ecological study included all session tasks, including warm-up.
-very low heart rate
Authors: Thank you for your note.
On this occasion, these results may be striking to the reader for the reasons stated above.
For this reason, the following paragraph has been included in the results section:
"As can be seen in Table 2 some variables present poor values for the load demands initially assumed to be made by professional basketball players. It is necessary to inform that in this first descriptive analysis all training tasks were included, including those performed during warm-up."
Discussion
The discussion is poorly presented and does not connect well to the used references in the introduction.
Authors: Thank you for your comments. The discussion has been increased and some references were added.
The authors have tried to correctly use some of the references included in the introduction, as well as new references. All the changes made can be seen in the discussion section.
You are talking about objectives but you didn't present the objectives of this trainings and the session results don't present any added value. To state that ''The coaching staff must know how to carry out the training process'' is like stating the obvious as the team was a professional team.
Authors: Thank you for your note. This information has been added and increased.
This information has been added and expanded in different paragraphs of the discussion.
Additionally, the following paragraph has been included in the conclusion:
“The monitoring of the players during the preseason makes it possible to follow the first training sessions of the players who make up the team after a period of inactivity. These records allow us to know the individual adaptations of the players to the demands of the tasks and to personalize the intensities of the tasks during the season.”
The limitations section is very short and should be extended.
Authors: Thank you for your perspective. This section has been extended.
Overall, in this condition the methods are poorly described and some results does not make sense. Therefore, I am rejecting this study.
Authors: Thank you for your comments. The authors take in account your comments and they have improved the document deeply. We hope that the improvements included will serve to change your decision.
Comments on the Quality of English Language
Extensive editing of the English language required
Authors: Thank you for your perspective. The authors have improved the English language.
Submission Date
12 July 2023
Date of this review
01 Aug 2023 03:23:29

Reviewer 3 Report
Influence of the task and the player position in the external and internal load in professional basketball players
This is a very interesting study in basketball, combining both internal and external indexes to evaluate the training load of players. I comment the Authors for using a high level basketball team which is a strong aspect of the study and should be highlighted in the limitations section. The study is well written, the methodology needs some more details and overall the manuscript is not very far from publication. We all know that the evaluation of training load is a significant factor for designing effective training programs and evaluate the player’s training-adaptations. Considering that basketball is a high intensity team sport, with physical contacts and powerful movements this study provides evidences about the evaluation of training load during basketball training.
General comment:
Authors evaluate the internal and external load during match stimulation which is not clearly presented in the methods. Someone may assume that resistance training, aerobic training or plyometrics may be used in the analysis. Lines 193-195 are clearly state that but Authors should present this earlier in the text.
Abstract:
Inside the manuscript Authors suppurate players into specific positions (see lines 103-104). Please, do not use other terms like Pivot since it might confuse readers. Correct this throughout the manuscript (line 212).
Abstract is too general. Maybe adding the type of tasks as presented in table 1 (Cinematics EL, Neuromuscular EL and Objective IL).
Introduction
Intro is well written and provides a good overview of the scientific literature regarding the presentation and connection of training load with performance. However, I missed the research questions that may lead to the study. For example, Authors focus on the importance of evaluating the training load during pre-season. Why this is important? And further to that, why coaches and strength and conditioning professionals should monitor training load during pre-season and what they should change during the competitive period?
Introduction is very good; just need some modifications to connect with the research question.
Materials and Methods
Participants: Authors stated that players were members of a professional men's basketball team of the highest Spanish category. More details about players should be provided and more specific: Training experience, number of players in each position, training sessions per week, inclusion and exclusion requirements to participate in the study etc.
Table 1 present the depended variables. Why resistance training is not evaluated? Moreover, are these variables referring only to the actual basketball training?
Exponent the 2 in the m/sec2.
Line 111: Authors should clarify the “different training sessions” for readers.
Are the instruments evaluated for ICCs? Is this methodology been used again in previous studies?
Results
Is table 2 presents the total amount of training load for all training sessions?
Table 3 and 4: Define the abbreviation of SSG. I am not sure that “Equality SSG” has been presented in the text in order to provide an abbreviation.
Discussion
Authors stated that “…36% of the tasks dedicated to the Full 181 Game have a duration greater than 16.22 minutes…”. I suggest to Authors adding some examples of these tasks.
Similar for “91.2% 183 of the tasks dedicated to individual work have a short or very short duration, i.e., they do 184 not exceed 12 minutes of work…. “
Lines 209-210: Authors should highlight the importance of pre-season on player’s performance during the in-season.
Conclusions are great.
Good job, well done.
Author Response
Reviewer 3
Open Review
( ) I would not like to sign my review report
(x) I would like to sign my review report
Quality of English Language
( ) I am not qualified to assess the quality of English in this paper
( ) English very difficult to understand/incomprehensible
( ) Extensive editing of English language required
( ) Moderate editing of English language required
( ) Minor editing of English language required
(x) English language fine. No issues detected
|
Yes |
Can be improved |
Must be improved |
Not applicable |
|
|
Does the introduction provide sufficient background and include all relevant references? |
(x) |
( ) |
( ) |
( ) |
|
Are all the cited references relevant to the research? |
(x) |
( ) |
( ) |
( ) |
|
Is the research design appropriate? |
(x) |
( ) |
( ) |
( ) |
|
Are the methods adequately described? |
( ) |
(x) |
( ) |
( ) |
|
Are the results clearly presented? |
(x) |
( ) |
( ) |
( ) |
|
Are the conclusions supported by the results? |
(x) |
( ) |
( ) |
( ) |
Comments and Suggestions for Authors
Influence of the task and the player position in the external and internal load in professional basketball players
This is a very interesting study in basketball, combining both internal and external indexes to evaluate the training load of players. I comment the Authors for using a high level basketball team which is a strong aspect of the study and should be highlighted in the limitations section. The study is well written, the methodology needs some more details and overall the manuscript is not very far from publication. We all know that the evaluation of training load is a significant factor for designing effective training programs and evaluate the player’s training-adaptations. Considering that basketball is a high intensity team sport, with physical contacts and powerful movements this study provides evidences about the evaluation of training load during basketball training.
Authors: Thank you for your perspective.
General comment:
Authors evaluate the internal and external load during match stimulation which is not clearly presented in the methods. Someone may assume that resistance training, aerobic training or plyometrics may be used in the analysis. Lines 193-195 are clearly state that but Authors should present this earlier in the text.
Authors: Thank you for your note. The aim of this study is to analyse the internal and external load of basketball players during two microcycles of training in the pre-season. This has been explained deeply in the methods section (lines 95-109).
Abstract:
Inside the manuscript Authors suppurate players into specific positions (see lines 103-104). Please, do not use other terms like Pivot since it might confuse readers. Correct this throughout the manuscript (line 212).
Authors: Thank you for your comments. There was a mistake in the variables section, in which the specific positions are specified. That was corrected and all the terms have been unified, in order not to confuse the readers.
Abstract is too general. Maybe adding the type of tasks as presented in table 1 (Cinematics EL, Neuromuscular EL and Objective IL).
Authors: Thank you for your note.
This information has been added to the abstract, despite the word limitation (200 words).
The following sentence has been included in the summary:
“Seventeen load variables were analyzed, and organized into kinematics external load, neuromuscules external load, and internal load. All variables were normalized to the same time unit (minute).”
Introduction
Intro is well written and provides a good overview of the scientific literature regarding the presentation and connection of training load with performance. However, I missed the research questions that may lead to the study. For example, Authors focus on the importance of evaluating the training load during pre-season. Why this is important? And further to that, why coaches and strength and conditioning professionals should monitor training load during pre-season and what they should change during the competitive period?
Introduction is very good; just need some modifications to connect with the research question.
Authors: Thank you for your comments.
The following paragraph has been added before the statement of the objectives of the study:
"The monitoring of training tasks is very important as it allows coaches to know the load demands that the players endure. This process is fundamental during the preseason, as the first training sessions of the players that make up the team (new players and veteran players) are monitored after a period of inactivity. These records allow the coaches to know the individual responses of the players and the adaptation to the new demands of the tasks. Once the players' responses have been identified, coaches can personalize task intensities during the season. Therefore, it is necessary to know what the players' responses are like during this first period of training."
Materials and Methods
Participants: Authors stated that players were members of a professional men's basketball team of the highest Spanish category. More details about players should be provided and more specific: Training experience, number of players in each position, training sessions per week, inclusion and exclusion requirements to participate in the study etc.
Authors: Thank you for your comments.
The following information has been added to the participants section:
“The sample was distributed as point guard (n=5), shooting guard (n=2), forward (n=4), power forward (n=1), and center (n=3). The team developed 6 training sessions per week with an average duration of 2h per session. The aim of these sessions was to im-prove technical and tactical skills of the players through different tasks.
In order to select the sample participants of the study, some inclusion criteria were established: i) the player must have completed at least 90% of the training sessions developed, ii) officially be part of the team at the beginning of the pre-season, and iii) not to present any injuries during this period.”
Table 1 present the depended variables. Why resistance training is not evaluated? Moreover, are these variables referring only to the actual basketball training?
Authors: Thank you for your perspective. The author used de WIMU device to quantify the physical fitness of the players through the analysis of the variables shown in table 1. No specific monitoring of endurance training was carried out, as the team's technical staff did not design specific training sessions for this physical quality outside the court. All training sessions were conducted on the court, except for the specific strength training sessions.
Exponent the 2 in the m/sec2.
Authors: Thank you for your note. This mistake has been corrected.
Line 111: Authors should clarify the “different training sessions” for readers.
Authors: Thank you for your comments.
The following information has been added to the procedure and instruments section:
“During this period, the players were monitored during the different training sessions carried out during this time, in order to analyze the different tasks developed. All the tasks designed by the coaching staff during the technical-tactical training carried out on the court were analyzed. The specific sessions of shooting training were not analyzed, as well as the specific sessions of strength training carried out in the gymnasium.”
Are the instruments evaluated for ICCs? Is this methodology been used again in previous studies?
Authors: Thank you for your note. These devices have been used in a great number of studies to analyse the external and internal load in different sports. Also, has been evaluated for ICCs.
Results
Is table 2 presents the total amount of training load for all training sessions?
Authors: Thank you for your note. Yes, Table 2 shows the information related to the totality of sessions.
Table 3 and 4: Define the abbreviation of SSG. I am not sure that “Equality SSG” has been presented in the text in order to provide an abbreviation.
Authors: Thank you for your comments.
The following clarification has been made in the variables section:
Small Sided Game of numerical equality (SSGe), Small Side Game of numerical inequality (SSGi).
Discussion
Authors stated that “…36% of the tasks dedicated to the Full 181 Game have a duration greater than 16.22 minutes…”. I suggest to Authors adding some examples of these tasks.
Authors: Thank you for your comments.
This information has been added in the text:
"5 vs 5 game simulation conducted in half or full field, in which the coach explains and introduces the team's attacking and defensive tactical concepts. "
Similar for “91.2% 183 of the tasks dedicated to individual work have a short or very short duration, i.e., they do 184 not exceed 12 minutes of work…. “
Authors: Thank you for your comments.
This information has been added.
“, such as technical tasks for warm-up, layup, or dribble, free throws, individual technique circuits, etc.,
Lines 209-210: Authors should highlight the importance of pre-season on player’s performance during the in-season.
Authors: Thank you for your comments. This information has been added.
Conclusions are great.
Authors: Thank you for your perspective.
Good job, well done.
Authors: Thank you for your comments.
Submission Date
12 July 2023
Date of this review
09 Aug 2023 11:21:09

Round 2
Reviewer 2 Report
Dear Authors
Thank you for addressing all of my questions and suggestions. The paper's quality improved. In my opinion, the paper is ready to be published in its current form.
Kind regards
Minor editing of English language required
Author Response
Thank you so much
Reviewer 3 Report
No comments.
Author Response
Thank you so much